# Evaluation of the Cleaning Effect of Natural-Based Biocides: Application on Different Phototropic Biofilms Colonizing the Same Granite Wall

Chiara Genova [1],* , Elsa Fuentes [2], Gabriele Favero [3] and Beatriz Prieto [2]

1 Department of Chemistry, Sapienza University of Rome, P.le Aldo Moro 5, 00185 Rome, Italy
2 Departamento de Edafoloxía e Química Agrícola, Facultade de Farmacia, Universidade de Santiago de Compostela, 15782 Santiago de Compostela, Spain
3 Department of Environmental Biology, Sapienza University of Rome, P.le Aldo Moro 5, 00185 Rome, Italy
* Correspondence: chiara.genova@uniroma1.it

**Abstract:** Natural derivatives, such as essential oils, are presented as an alternative to classical biocides to the treatment of biocolonization. Thus, in this work, the cleaning and biocidal potential of some natural derivatives towards two natural biofilms' growth on the same granite wall, with different microbial composition, was evaluated. For this purpose, three essential oils (EOs) (from *Origanum vulgare*, *Thymus vulgaris* and *Calamintha nepeta*) and their main active principles (APs) (carvacrol, thymol and R-(+)-pulegone, respectively) were embedded in a hydrogel matrix, with different combinations of EOs and APs, in order to evaluate the synergistic action of different actives. For comparative purposes, pure hydrogel and a mechanical method (brushing) were also used. Colorimetric measurements and chlorophyll *a* fluorescence analyses were performed to evaluate the cleaning action of the treatments on the biofilms. Overall, the EOs and APs present in the hydrogel proved to be reliable treatments to limit natural biocolonization, with *O. vulgare* being one of the most effective treatments in combination with other compounds, due to the majority presence of carvacrol. Moreover, the effect of the different treatments strictly depended on the biofilm in question, as well as its ability to adhere to the substrate.

**Keywords:** essential oils; active principles from essential oils; granite; phototropic biofilms; green products; conservation of cultural heritage; non-invasive measurements





## 1. Introduction

In recent years, the concept of sustainability has been steadily garnering more attention in all research fields, and the field dealing with the development of new products for various applications aimed at preserving artistic artifacts is not excluded.

Indeed, a new definition for the green conservation of cultural heritage was recently suggested as "all the eco-sustainable practices to be used in the conservation and restoration of Cultural Heritage assets, as alternatives to traditional products and methods which are often toxic and harmful for the users and the environment" [1]. In this scenario, one of the main goals for scientists is finding alternative eco-friendly solutions to classical biocides to prevent and eliminate biopatinas from porous materials that are inherently susceptible to biological attack, as in the case of mineral building materials. The intense interest in this subject is clear, since biodeterioration is considered one of the main expenses and widespread problems for craftsmen and building owners [2], as it was estimated that over 20%–30% of stone deterioration is caused by microorganisms [3], responsible for irreversible chemical, physical and aesthetic damage both on surfaces and in the inner structure of the porous matrix.

The investigation into new biocidal solutions is not only aimed at finding strategies to reduce a possible environmental impact, but also to limit other drawbacks shown by the methods used up to now to remove biofilms from a stone's surface.

This is the case for chemical biocides based on isothiazolinones (ITs) and quaternary ammonium salts (QACs) [2,4], which have been widely employed thanks to their well-known biocidal properties but present some incompatibilities with the composition of surfaces and biofilms. For instance, in a study by Sanmartin et al. (2020) [5], it was observed that the application of some commercial QAC-based biocides (i.e., Biotin R®, Biotin T®, Preventol RI80® and New Des 50®), due to their abrasive nature, can produce a slight sanding of surfaces, leading to a possible worsening of the mechanical damage [6]. In the same study, it was observed that a long-term biocidal effect does not occur for all the biocides employed, and it was also assessed that New Des 50® is inefficacious towards *Apatococcus lobatus*, confirming that chemical biocides are not always selective towards specific biodeteriogens, as also occurs for IT-compounds towards cyanobacteria [7]. Furthermore, in some remarkable circumstances, as intensively investigated for the case of the Caves of Lascaux [8,9], the employment of benzalkonium chloride (a QAC derivative) induced the increase in organic carbon source on surfaces, promoting secondary recolonization by more harmful and biocidal-resistant microorganisms not originally detected.

An alternative is provided via mechanical and physical tools that assure the cleaning of surfaces without the employment of chemicals or other additional compounds. The mechanical removal of biopatina is a procedure commonly used in the past, which envisages the use of brushes, scalpels, water pressure washers, vacuum cleaners, and abrasive systems [2,4]. However, mechanical cleaning is not selective and does not prevent recolonization but, more relevantly, can lead to a worsening of the damage, causing (a) detachments of minerals and other inorganic fragments from surfaces and (b) a deeper penetration of the microorganisms in the inner structure of the porous matrix. In terms of physical methods, lasers and other devices based on the emission of energetic beams (for example, γ- radiation and UV-C irradiation) were also employed to remove superficial biopatinas. In addition to high costs, these instruments require specific expertise to be handled and, since they do not prevent recolonization, their repeated use can lead to a further deterioration of the irradiated surfaces [2,4].

A promising sustainable alterative to the systems previously described is represented by natural extracts having intrinsic biocidal properties, among which are essential oils (EOs). Essential oils are volatile, organic compounds extracted from various plants and plant materials, assessed as having broad-spectrum bactericidal, viricidal and fungicidal properties [10]. They are considered safer than classical biocides and are classified as GRAS (Generally Recognized As Safe) according to the FDA (Food and Drug Administration) guidelines [11]. Considering the above, and because their assessed biocidal action occurs at very low concentrations [12], in recent years there also has been an increased interest in testing these substances against biodeteriogens of stone materials, as evidenced by recent papers and reviews dealing with this subject [1,2,4,13], in which the strength and weak points of EOs have been discussed. In this sense, one of the most controversial aspects regards the impossibility of reproducing the same experimental conditions to realize a biocidal product based on EOs [10,14] due to their very heterogeneous chemical composition, which is governed by their natural origin and is affected by numerous external agents, such as harvesting time, chemical properties of the soil, the part of the plant employed, sun exposure as well as the influence of the other environmental agents [14]. However, although EOs show a very complex chemical profile (60–300 chemicals) [15], it is well known that each oil is characterized by the prevalence of one or two active components at high concentrations [10], also establishing the chemotype. Several studies highlighted how the presence of main components influence their biocidal action [16], so much so that some authors refer to them as essential oils' active principles (APs) [17], and they can be considered possible substitutes of the oils to prevent cultural heritage biocolonization [18,19]. Finally, the employment of single active substances can reduce high production costs associated with the extraction of high-quality EOs [20].

Another aspect that must be considered is that both EOs and their main constituents are volatile compounds, and their short duration on surfaces could limit their biological

interactions with microorganisms [19,21]. For this reason, and because direct application of any conservation product is never recommended [22], a vehiculation system is required. Particularly suitable in this sense are polyvinyl alcohol (PVA)-based hydrogels, both for their "green" composition [23,24] and for the efficacy showed in the encapsulation of essential oils from different species (i.e., *T. vulgaris*, *L. angustifolia*, *O. vulgare* and *C. nepeta*) to be applied on biodeteriorated materials [25–29].

Considering that only recently some progress has been made regarding the application of such substances on site and on spontaneously grown biofilms [11,30–32] and that there are still not enough experiments to discuss the interference of the natural compounds with the heritage materials [4], in this study we investigated the potential of combined systems based on natural derivative biocides carried in a PVA-gellan hydrogel, for future applications on stone-made cultural heritage.

The natural biocides selected are three essential oils from *Origanum vulgare* L., *Thymus vulgaris* L. and *Calamintha nepeta* (L.) Savi (a synonym for *Clinopodium nepeta* (L.) Kunze [33]). Furthermore, to evaluate the biocidal contribution of their main components, the APs from the respective EOs were also tested, and they are carvacrol for *O. vulgare*, thymol for *T. vulgaris* and R-(+)-pulegone for *C. nepeta*.

The three EOs were chosen for their assessed biocidal activities, and they belong to three common Mediterranean plants (viz. oregano, thyme and pennyroyal) from the Lamiaceae family. In particular, *O. vulgare* and *T. vulgaris* resulted to be the most studied and effective EOs towards several biodeteriogens of cultural heritage [13], and similar results were yielded for *C. nepeta* [34,35].

Despite the large-spectrum biocidal action shown by these substances, they can be active (or not) towards specific microorganisms. An example is reported in the study of Panizzi et al. (1993) [36], where the EOs from *C. nepeta* and *T. vulgaris*, compared to other EOs (*Satureja montana* L., *Rosmarinus officinalis* L), demonstrated the most powerful inhibitory action towards several bacteria and mycetes strains. However, only *C. nepeta* was effective against the most resistant bacterium (*Pseudomonas aeruginosa*), contrary to *T. vulgaris*. For this reason, in this study, EOs and APs were employed both alone and combined, to assess (i) the action of the substances towards targeted biodeteriogens and (ii) an eventual synergism occurring between the chemicals.

The experimental set-up follows the one built for two previous research studies that provided for the employment of such systems on two lithotypes frequently used as built materials of cultural heritage, that are granite and travertine. In the first case, the products were applied on artificially biocolonized granite samples under controlled laboratory conditions [29], while, in the second case, the experimental procedures were performed on site on a biofilm naturally grown on travertine [28]. Compared to the abovementioned study cases, here we performed the in situ application of the products on two phototropic biofilms with different microbial compositions, spontaneously grown on a granite wall. Indeed, one of the aims proposed here is assessing the efficacy of the selected substances depending on the microorganism involved and the biofilm's characteristics.

As travertine was chosen in our previous study [28] as a representative artistic and historical material for the city of Rome [37], for the same reason, granite has been selected because its wide diffusion in Santiago de Compostela, where this study was carried out and where the most relevant historical and artistic sites are typically made of this lithotype [38,39]. What is reported above fits with our long-term goal, which foresees the ideation of a protocol for the in situ treatment of a wide range of lithotypes, taking into account the intrinsic characteristics of the treated stone (porosity, roughness, scratch resistance, etc.), but also the type of biofilm growth on it. Furthermore, in this case, we advanced our methodology by coupling colorimetric investigations, used to monitor cleaning efficacy immediately after the removal of the substances, with the assessment of the vitality of remaining microorganisms as an indicator of the biocidal effect of the biocides employed. It must be highlighted that, for the techniques employed, all are portable and non-invasive, in view of future applications on cultural heritage.

## 2. Materials and Methods

### 2.1. Essential Oils and Their Main Components

The experiment was aimed at assessing the suitability of biocides based on natural compounds to eliminate spontaneous biofilms colonizing an outdoor exposed granite wall. In particular, we tested three essential oils (EOs) from *Origanum vulgare* L., *Thymus vulgaris* L. and *Calamintha nepeta* (L.) Savi, purchased from specialized retailers and producers of natural extracts from plants. The EOs of *O. vulgare* and *T. vulgaris* were supplied from Esencias Martínez Lozano (Murcia, Spain) and the EO of *C. nepeta* from Joulienne Fauconnier (Corsica, France). Because we also wanted to compare the effect of EOs with the one of their main components, or active principles (APs), pure carvacrol (≥98%), thymol (≥98.5%) and R-(+)-pulegone (≥90%) were obtained from Sigma-Aldrich (St. Louis, MO, USA).

### 2.2. Experimental Set Up

For the experiment, sixteen treatments that included the EOs, their APs, a PVA gellan-based hydrogel and a mechanical method (brushing) were employed for the removal of biofilms from an outdoor exposed granite wall (Faculty of Pharmacy, Campus Vida, University of Santiago de Compostela, Galicia, Spain) showing evident biocolonization.

In particular, the wall apparently showed two types of biocolonization. This assumption arises from observed differences in color occurring between two different zones of the granite surface (Surface a and Surface b), as reported in Figure 1.

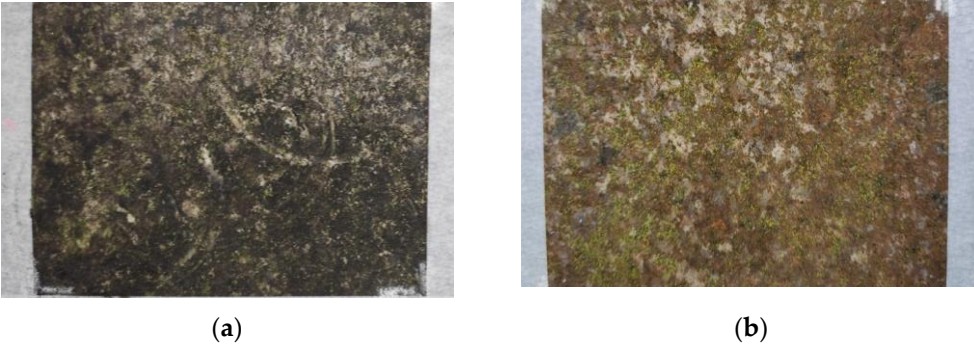

(**a**)          (**b**)

**Figure 1.** Experimental surfaces (details). In the pictures, the two surfaces are compared. In (**a**) the dark green biopatina is shown (Surface a), and in (**b**) the green-orange biopatina is shown (Surface b).

The surface identified as Surface a showed a visible and thick biopatina, characterized by a darker coloration, on the other side; Surface b showed a patina less uniformly distributed and characterized by a lighter green-orange coloration.

The choice of testing biocidal products against two biopatinas with a different microbial composition was performed to assess the possible difference in the efficacy of the biocides against target microorganisms.

To this aim, both surfaces were divided into sixteen squares (dim 10 cm x 10 cm), each of them subjected to a different treatment, identified with the letter T (from T1a to T16a for Surface a and from T1b to T16b for Surface b) and presented in more detail in Figure 2. In particular, from T1 to T14, the treatments contain EOs and APs, alone and blended, and they are composed as follows: two groups containing a single EO (T1 = *O. vulgare*, T2 = *T. vulgaris*, T3 = *C. nepeta*) and a single AP (T4 = carvacrol, T5 = thymol, T6 = pulegone); two groups composed by a combination of two EOs (T7 = *O. vulgare* + *T. vulgaris*, T8 = *O. vulgare* + *C. nepeta*, T9 = *T. vulgaris* + *C. nepeta*) and the combination of two APs (T11 = carvacrol + thymol, T12 = carvacrol + pulegone, T13 = thymol + pulegone); one treatment containing the combination of three EOs (T10 = *O. vulgare* + *T. vulgaris* + *C. nepeta*) and another containing a combination of the three APs (T14 = thymol + carvacrol + pulegone).

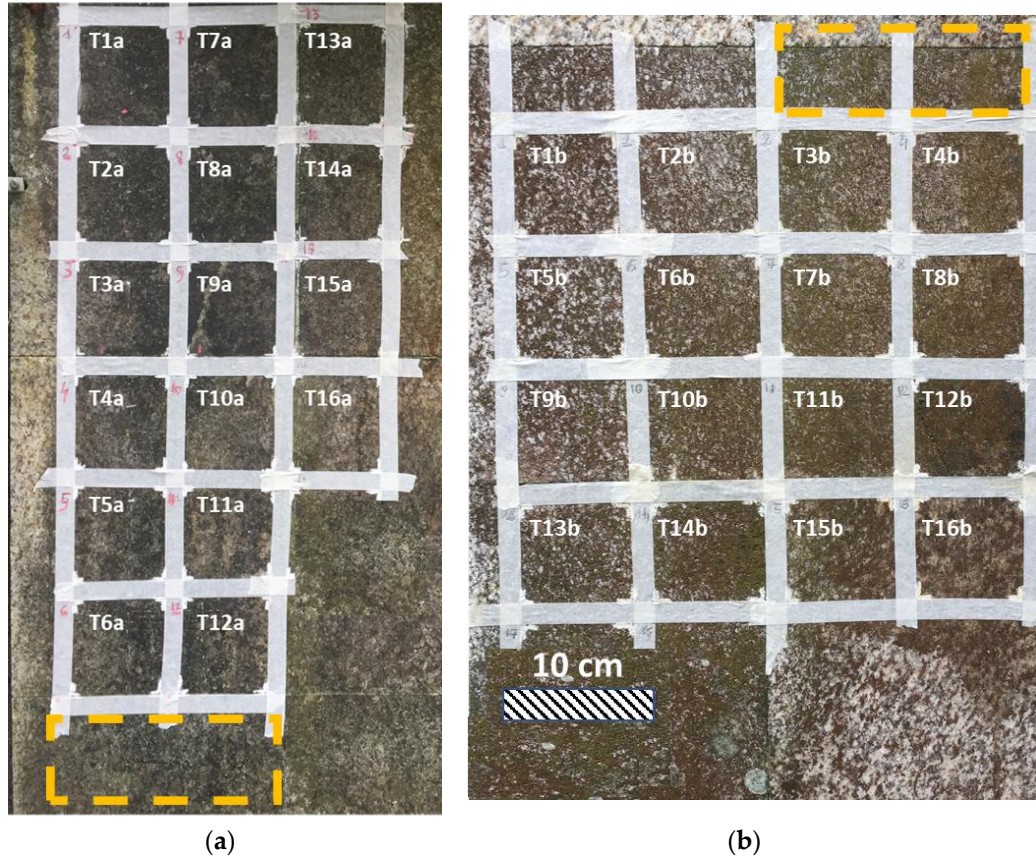

**(a)**                                    **(b)**

**Figure 2.** (**a**) Surface a; (**b**) Surface b. Composition of each treatment: T1 = *O. vulgare*; T2 = *T. vulgaris*; T3 = *C. nepeta*; T4 = Carvacrol; T5 = Thymol; T6 = Pulegone; T7 = *O. vulgare + T. vulgaris*; T8 = *O. vulgare + C. nepeta*; T9 = *C. nepeta + T. vulgaris*; T10 = *O. vulgare + T. vulgaris + C. nepeta*; T11 = Carvacrol + Thymol; T12 = Carvacrol + Pulegone; T13 = Pulegone + Thymol; T14 = Carvacrol + Thymol + Pulegone; T15 = Hydrogel; T16 = Brushing. The orange rectangles individuate the zones where the biological material was collected.

For their application, all the substances were loaded into a PVA gellan-based hydrogel crosslinked with CaCl$_2$ and enriched with a surfactant, already used and described in previous works [28,29], useful for vehiculating the natural biocides on surfaces. The formulations were realized by adding a calculated amount of EO or AP to the hydrogel and stirring at room temperature for 15 min with a magnetic stirrer without heating, until the formation of emulsions.

The treatments were left to act for one month, and then removed. Pure hydrogel (T15) and mechanical brushing with a hard brush on a humid surface (T16) were also used. All the treatments, the compositions and the overall concentrations are schematically shown in Figure 3. However, the method used and the concentrations of each EO and APs present in the formulations, as results of the experiment, are described in detail in Section 3.2.

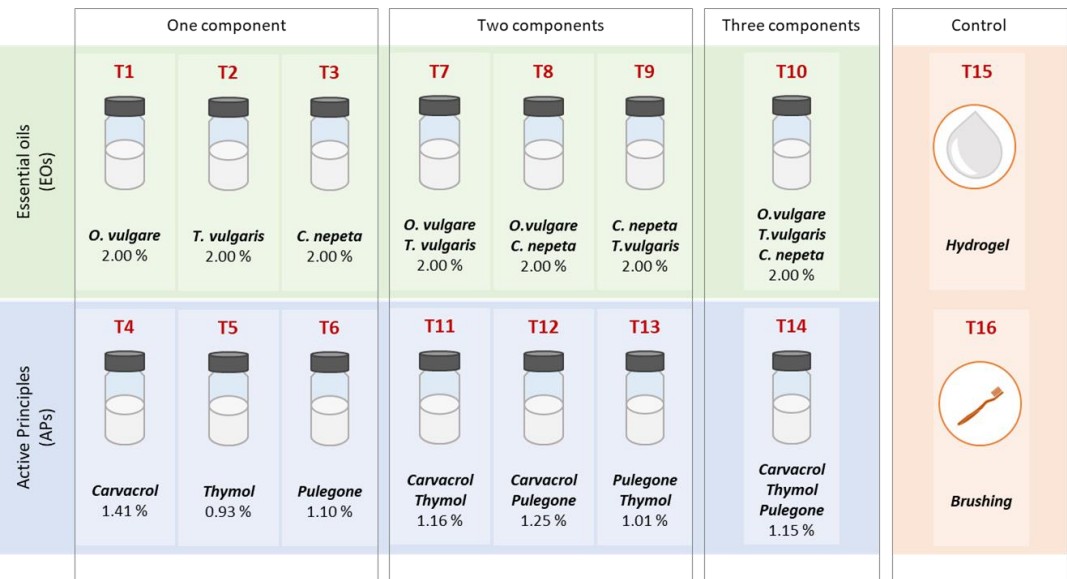

**Figure 3.** Schematic representation of the treatments employed and their composition.

### 2.3. Biofilm Sampling and Taxonomic Characterization

Two areas of the wall adjacent to the two experimental surfaces, showing similarities in type and degree of biological colonization, were selected to collect the biofilms (Figure 2). The sampling was performed in a non-invasive way by using a swab that was gently rubbed on the biocolonized surfaces and then stored in sterile tubes containing a buffer solution made of 5% of NaCl. The sampling was repeated in triplicate for both the biofilms present on Surface a and Surface b. The biological material was morphologically characterized by microscopic observations and was examined under light microscopy (LM) Nikon Eclipse E600 (Nikon Instruments, Melville, NY ) equipped with an E-Plan 40× objective (N.A. 0.65) and differential interference contrast (Nomarski) optics. LM photographs were taken with an AxioCam ICc5 Zeiss digital camera (Zeiss, Oberkochen, Germany ). The species identification and nomenclature used was mainly based on the following: Rindi (2011) [40] and Škaloud (2018) [41] for the identification of green algae (Chlorophyceae); Komárek (1930) [42] and (2016) [43] for cyanobacteria (Cyanoprokariota); and Lange-Bertalot and Hoffman (2011) [44] for diatoms (Bacillariophyceae). The samples were homogenized and the taxa cells quantitatively counted in a Utermohl sedimentation chamber (Utermöhl 1958) [45]; the abundances of taxa were expressed in a percentage from a total count of 1000 cells.

### 2.4. Colorimetric Investigations

Color measurements have been demonstrated to be a rapid, efficient and reliable non-invasive and non-destructive tool to assess the presence of and quantify phototrophic biofilms on a stone's surface. Colorimetric characterizations, indeed, have been employed in the past to monitor the degree of colonization on surfaces [46,47], to assess the efficacy of biocidal treatments on a stone's materials over time [5], as well as to estimate the cleaning effects of different products in removing superficial biofilms from surfaces [29].

The apparatus employed for the colorimetric measures is a portable spectrophotometer (CR-310, Konica Minolta, Japan) equipped with a measure head with a 50-mm-diameter viewing area Illuminant D65, 2° observer; specular component included (SCI) mode was employed as measurements conditions. Colorimetric acquisitions on three randomly selected points were performed for each humid surface, and then the mean value was calculated and considered representative of the overall color of the analyzed area. The Cartesian colorimetric parameters a*(changes in redness-greenness) and b* (changes in blueness-yellowness) were employed to detect microbial colonization, since they already

proved to be relevant indicators of pioneer microorganisms (algae and cyanobacteria) forming biofilms on rock surfaces [48].

The measures were performed at the beginning of the experimental procedure ($t_0$), or on the biocolonized surfaces, and immediately after the removal of the treatments to assess their cleaning efficacy ($t_1$). Further measures were realized on an uncolonized part of the same wall, used as a reference of the original color of granite.

Color data were plotted in a bidimensional ab* Cartesian plane, and the evolution of the color between $t_0$ and $t_1$ was evaluated for each treatment and compared with the color of the uncolonized granite. To this end, partial color differences $\Delta a*$ and $\Delta b*$ were calculated to assess variations in color between surfaces after the treatments and the uncolonized granite. Positive values of $\Delta a*$ indicate reddening, and negative values indicate greening. Positive values of $\Delta b*$ indicate yellowing, and negative values indicate bluing.

### 2.5. Pulse Amplitude Modulated (PAM) Fluorometry

To assess the biological activity of the microorganisms, chlorophyll *a* (chll a) fluorescence measurements were performed at $t_0$ and $t_1$ to support the colorimetric investigations for the biofilm characterization and for the assessment of cleaning and biocidal effects of the treatments. A pulse amplitude modulated (PAM,) fluorometry apparatus was employed, a Phyto-PAM (Heinz Walz GmbH, Effeltrich, Germany) equipped with a fiberoptic emitter-detector unit Phyto-EDF, for the acquisition of fluorescence signals at 665 nm.

The apparatus requires darkness conditions to obtain a signal, to guarantee an ample dark-adaptation time and to allow the full oxidoreduction state of the PSII reaction centers [49]. For this reason, a black plastic cover was laid on the surfaces at least 20 min before the acquisitions. Measurements were performed under the cover at hours of the day when the surfaces were not directly exposed to sunlight [5].

For each panel, five readings were acquired on five randomly selected points, and then the average value, representative of one panel, was calculated.

Signals recorded at 665 nm are the minimal fluorescence signal of dark-adapted cells ($F_0$) and the maximal fluorescence signal after a saturating light pulse in dark-adapted cells ($F_m$). These parameters made it possible to determinate the maximum quantum yield ($Y = F_v/F_m = (F_m - F_0)/F_m$), which is an indicator of the viability of photosynthetic microorganisms [50]. Relative differences $\Delta Y = Yt_0 - Yt_1$ were calculated as an estimator of the vital activity of microorganisms, where $\Delta Y > 0$ establishes the increase in the vital activity after the application of the treatments and, conversely, $\Delta Y < 0$ the decrease in the vital activity.

### 2.6. Quantification of Extracellular Polymeric Substances (EPS)

The carbohydrate component of EPS was extracted following the method described by Yang et al. (2019) [51], with some modifications. The biofilm was scraped from the wall in three different areas (of each of the biofilms studied) and homogenized, and 0.05 g were collected and resuspended in 2 mL of NaCl. Samples were vortexed and sonicated for 2 min. After that, they were shaken horizontally at 150 rpm for 10 min and sonicated for another 2 min. The samples were then centrifuged for 15 min at 5000g. To separate the cell debris from the supernatant containing the EPS, samples were filtered through 0.45 μm nitrocellulose membranes (Millipore). The supernatant containing the LB-EPS (loosely bound EPS) was retained. The carbohydrate content of the EPS was measured using the phenol-sulfuric method [52] with glucose as standard.

### 3. Results

### 3.1. Characterization of the Biofilms

The biodiversity of the two biofilms is confirmed by taxonomical and morphological analyses (Figure 4), and the taxa composition and relative abundances data of microorganisms present are summarized in Table 1.

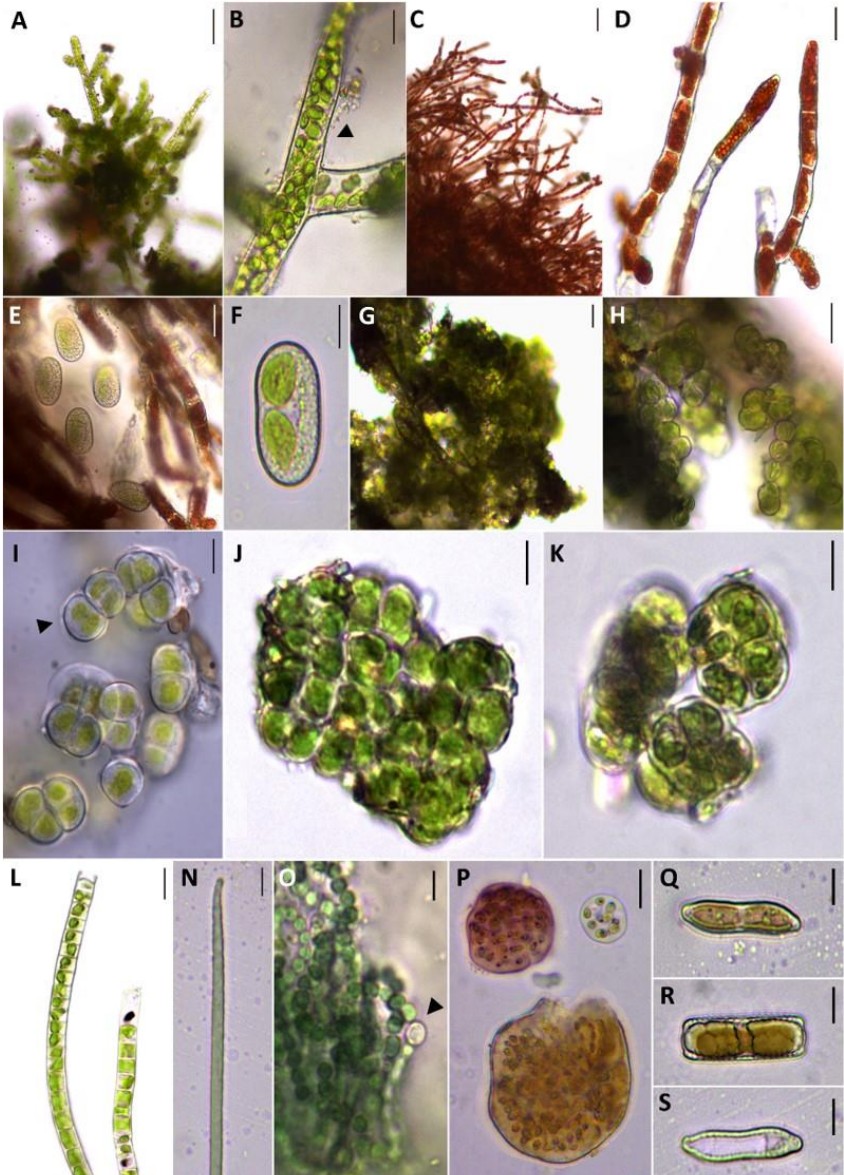

**Figure 4.** Taxa identified in sampled subaerial biofilms: (**A**,**B**) protonema of bryophyta; (**C**,**D**) *Trentepohlia aurea* (Linnaeus) C. Martius; (**E**,**F**) *Mesotaenium macrococcum* (Kützing ex Kützing) J. Roy & Bisset; (**G**–**I**) *Apatococcus lobatus* (Chodat) J.B.Petersen: diagnostic detail of the cells with bilobate chloroplast without pyrenoid indicate by arrow (**I**);(**J**,**K**) *Desmococcus olivaceus* (Persoon ex Acharius) J. R. Laundon; (**L**) *Klebsormidium flaccidum* (Kützing) P. C. Silva, K. R. Mattox & W. H. Blackwell; (**N**) *Oscillatoria formosa* Bory ex Gomont; (**O**) *Nostoc* sp. Vaucher ex Bornet & Flahault trichomes without mucilage formed by subspherical cells of 2.0-4.5 μm in diameter and heterocytes indicated by an arrow; (**P**) *Gloeocapsa punctata* Nägeli; (**Q**–**S**) *Hantzschia amphioxys* (Ehrenberg) Grunow. Scale bar = 50 μm (**A**,**C**,**G**); 20 μm (**E**,**H**); 10 μm (**B**,**D**); 5 μm (**F**,**I**,**K**,**L**–**S**).

**Table 1.** Composition and relative abundance of the subaerial biofilm taxa studied in the samples collected from Surface a (A1, A2, A3) and from Surface b (B1, B2, B3).

| Taxa | Phylum/Class * | Surface a | | | Surface b | | |
|---|---|---|---|---|---|---|---|
| | | A1 | A2 | A3 | B1 | B2 | B3 |
| Bryophyte (protonema) | Bryo | 0 | 0 | 0 | 8.1 | 0 | 5.6 |
| *Trentepohlia aurea* | Chloro | 0 | 0 | 0 | 73.7 | 75.2 | 71.6 |
| *Mesotaenium macrococcum* | Chloro | 0 | 2.5 | 0 | 5.8 | 12.8 | 9.7 |
| *Apatococcus lobatus* | Chloro | 51.6 | 37.8 | 36.2 | 6.4 | 9.2 | 8.5 |
| *Desmococcus olivaceus* | Chloro | 8.5 | 7.0 | 9.2 | 2.7 | 1.8 | 3.6 |
| *Klebsormidium flaccidum* | Chloro | 1.8 | 2.5 | 2.7 | 0 | 0 | 0 |
| *Oscillatoria Formosa* | Cyano | 0 | 1 | 0 | 1 | 1 | 1 |
| *Nostoc* sp. | Cyano | 12.5 | 9.5 | 10.1 | 0 | 0 | 0 |
| *Gloeocapsa punctata* | Cyano | 24.6 | 38.7 | 40.8 | 2.3 | 0 | 0 |
| *Hantzschia amphioxys* | Bac | 1 | 1 | 1 | 0 | 0 | 0 |
| | Bryo | 0 | 0 | 0 | 8.1 | 0 | 5.6 |
| | Chloro | 61.9 | 49.8 | 48.1 | 88.6 | 99.0 | 93.4 |
| | Cyano | 37.1 | 49.1 | 50.9 | 3.3 | 1 | 1 |
| | Bac | 1 | 1 | 1 | 0 | 0 | 0 |

* Bryo = Bryophyta; Chloro = Chlorophyceae; Cyano = Cyanoprokariota; Bac = Bacillariophyceae.

Both sub-aerial biofilms are mainly composed of green algae. Cyanobacteria have been also detected but are much more abundant in the samples collected from Surface a, which is also the one presenting the most heterogeneous microbial composition. This one, indeed, shows a co-dominance of green algae (48.1–61.9%) and cyanobacteria (37.1–50.9%), where they are dominating the taxa *Apatococcus lobatus* (Chodat) J.B.Petersen, *Desmococcus olivaceus* (Persoon ex Acharius) J.R.Laundon, *Nostoc* sp. Vaucher ex Bornet & Flahault, and *Gloeocapsa punctata* Nägeli (Table 1). On the other side, samples collected from Surface b are mainly composed of green algae (88.6–99.0%), especially dominating the *Trentepohlia aurea* (Linnaeus) C. Martius biomass (Table 1).

A different microbial composition also produced differences in the content of extacellular polymeric substances (EPS) that are much more abundant in the samples collected from Surface a than in Surface b, as reported in Figure 5.

Colorimetric analyses also establish differences in the distribution in the a*b* chart (see subfigure (a,b) of figure in Section 3.3) of color points from biocolonized ($t_0$) Surface a and Surface b. This evidence is in accordance with the different composition of biofilms, due to the production of colored pigments by specific microorganisms, as hypothesized at the beginning of the experiment.

In subfigure (a,b) of figure in Section 3.3, the color of the biocolonized surfaces is also compared with the reference uncolonized granite.

Chromatic color data from Surface a at $t_0$ are all included in the part of the chart characterized by negative values of a* and positive values of b* (subfigure (a) of figure in Section 3.3). This means that, in general, the green and yellow chromatic components predominate on the red and blue ones, respectively. Considering the distribution of the chromatic color data along the a*-axis at $t_0$, all the points are included in a small color gamut ranging from −3.6 to −1.2 CIELAB units. Conversely, the b* parameter varies more in the distribution, ranging from 5.3 and 10.9 CIELAB units along the b*-axis. Compared to

the biocolonized Surface a, the uncolonized granite (a* = −2.2, b* = 14.6) appears yellower (in all cases) and greener, except for T13a, which is the point showing the smaller value of a* (-3.6 CIELAB units).

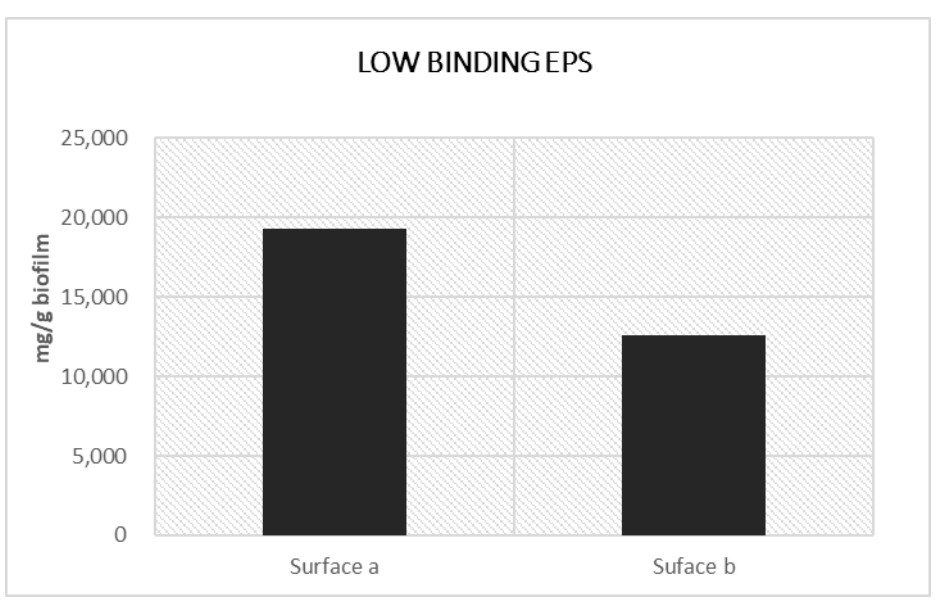

**Figure 5.** LB-carbohydrates present in biofilm patinas from Surface a and Surface b.

Color points from Surface b at $t_0$ show a wider distribution inside the colorimetric chart (subfigure (b) of figure in Section 3.3) than the points from Surface a, both along the a*-axis (min = −3.7, max = 0.3 CIELAB units) and the b*-axis (min = 14.9, max = 9.1 CIELAB). Compared to the data from Surface a, Surface b shows a predominance of the yellow component on the blue one, and, in some cases (T1b, T2b, T4b, T5b, T9b) also appears redder. This analysis is in accordance with biofilms' composition where the presence of microorganisms from the *Trentepohlia* species in Surface b is associated with a large production of carotenoids, especially β-carotene (followed by zeaxanthin, neoxanthin, lutein, ascorbic acid and α-tocopherol), which protects them from energetic UVa and UVb solar radiation. These photosynthetic pigments give these terrestrial algae a typical yellow-orange and red coloration, in accordance with what can be observed by the naked eye [39,53,54].

Nevertheless, it must be said that compared to Surface a, in many circumstances, the color spots from Surface b at $t_0$ are closer to the uncolonized reference.

This is not surprising, since the biofilm from Surface a also shows a high amount of EPS. In addition to providing mechanical stability to biofilms and an increase in the adhesion to the substrates, EPS entails the entrapment of substances variously dispersed in the atmosphere [55,56], contributing to the darker appearance of the biofilm. This is also favored by the presence of cyanobacteria *Nostoc* sp. and *Gloeocapsa* sp., already recognized as producers of blue-black colored pigmentation and biopatinas [57,58].

### 3.2. Method for the Realization of the Treatments

As previously described (Section 2.2), the application of the treatments provided the encapsulation of the EOs and APs in a PVA-gellan hydrogel matrix. Hydrogel, indeed, is a useful method for the vehiculation of the products on the surfaces, with the twofold function of reducing their volatility and allowing their efficient removal after cleaning. Furthermore, one of the advantages of our hydrogel consists in its ability to form a thick layer on surfaces that can be easily removed once dried (see following Figure 6), combining the biocidal action of the compounds towards microorganisms with an efficient removal of the superficial dirty layers, improving the cleaning effect.

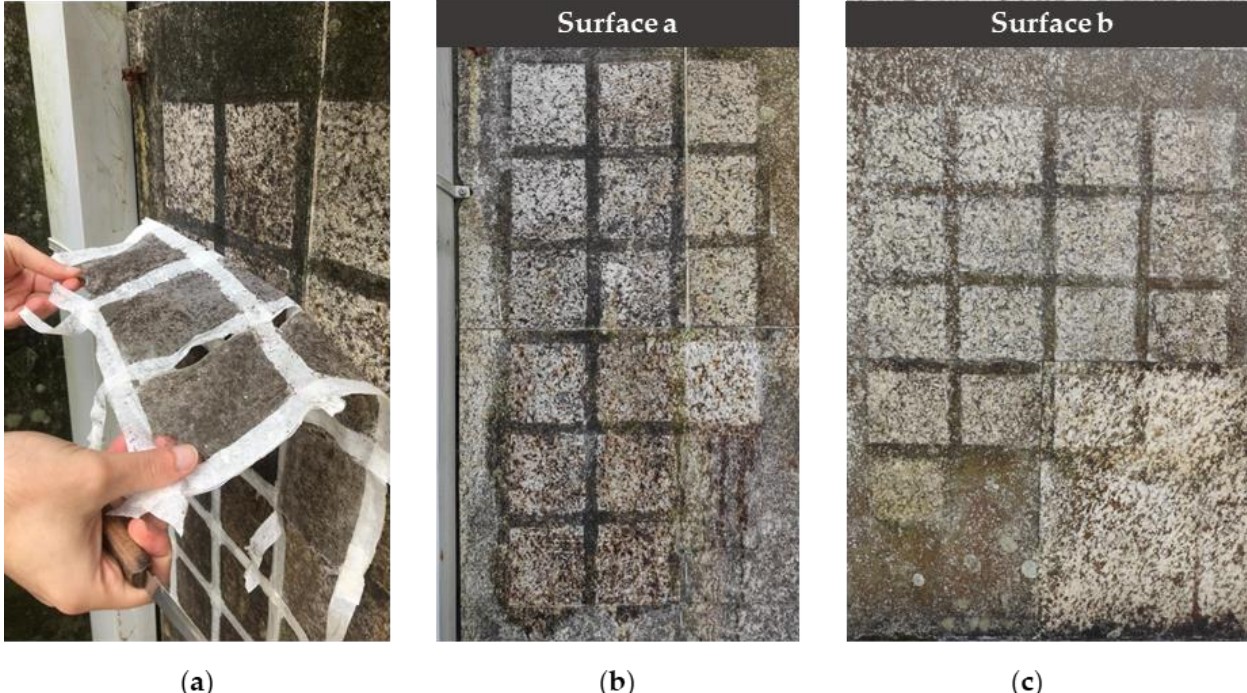

**Figure 6.** Appearance of the experimental surfaces after the cleaning procedures. (**a**) Peeling of the treatment that provided the employment of the hydrogel; (**b**) Surface a after the removal of the treatments ($t_1$) and (**c**) Surface b after the removal of the treatments ($t_1$).

All the treatments containing EOs alone (T1, T2, T3) have a concentration of 2% $w/w$, to obtain biocides having a large-spectrum action and maintaining a low ecological impact [59,60]. Moreover, it must be highlighted that the rheological and filming properties of the hydrogel are slightly modified in the presence of EOs and APs, reducing its viscosity and its adhesion on surfaces. However, a maximum concentration of 2% $w/w$ has already been demonstrated to be appropriate for such applications [28,29].

This concentration was also maintained for treatments containing the blend of two (T7, T8, T9) or three oils (T10), where the contribution of a single oil is, respectively, 1/2 and 1/3 of the overall concentration of 2% $w/w$.

As it concerns APs, since we wanted to assess the influence of the major components in the chemical composition of the EOs, the concentration of the single substances to realize the treatments reflects the one present in the respective essential oil. In particular, the chemical composition of the oils was already characterized in a previous work [29], and it was established that their main components are carvacrol for *O. vulgare* (70.5% of the total composition), thymol for *T. vulgaris* (46.4% of the total composition) and R-(+)-pulegone for *C. nepeta* (55.2% of the total composition).

Indeed, 2% $w/w$ was used as a reference to establish the weight of the AP that must be added to the hydrogel, and the calculated final concentrations are 1.14% $w/w$ of carvacrol in T4, 0.93% $w/w$ of thymol in T5 and 1.10% $w/w$ of R-(+)-pulegone in T6. The same evaluations are valid for treatments containing two (T11, T12, T13) or three (T14) APs combined.

All the concentrations and the composition of each treatment containing EOs and APs is reported in detail in the following Table 2.

**Table 2.** Chemical compositions and concentrations of EOs and APs present in each formulation.

| | Components [*w/w%*] | | | | | |
|---|---|---|---|---|---|---|
| Treatments | *O. vulgare* | *T. vulgaris* | *C. nepeta* | Carvacrol | Thymol | R-(+)-Pulegone |
| T1 | 2.00 | | | | | |
| T2 | | 2.00 | | | | |
| T3 | | | 2.00 | | | |
| T4 | | | | 1.41 | | |
| T5 | | | | | 0.93 | |
| T6 | | | | | | 1.10 |
| T7 | 1.00 | 1.00 | | | | |
| T8 | 1.00 | | 1.00 | | | |
| T9 | | 1.00 | 1.00 | | | |
| T10 | 0.67 | 0.67 | 0.67 | | | |
| T11 | | | | 0.70 | 0.46 | |
| T12 | | | | 0.70 | | 0.55 |
| T13 | | | | | 0.46 | 0.55 |
| T14 | | | | 0.47 | 0.31 | 0.37 |

### 3.3. Cleaning Efficacy of the Treatments

Naked-eye observations give preliminary evidence about the efficacy of all the treatments in removing the superficial biopatina, as shown in Figure 6.

In both cases, the main visual changes are given by mechanical brushing (T16a and T16b), which seems to be the treatment showing better results in the superficial biofilm's removal. For Surface b (Figure 6c), the hydrogel (T15b) also yielded similar results.

Colorimetric data obtained after cleaning (Figure 7c,d) return two different situations, depending on the considered surface.

Comparing the chromatic data from Surface a between $t_0$ and $t_1$, a general displacement of color points inside the a*b* chart can be observed at $t_1$ (Figure 7c).

Indeed, although the points at $t_0$ were more uniformly distributed (Figure 7a) than for Surface b (Figure 7b), and they were all included in the part of the plane characterized by positive values of b* and negative values of a*, at $t_1$ a different situation can be observed (Figure 7c,d). Considering the a* parameter after cleaning ($t_1$), the color points show a wider distribution along the a*-axis, so much so that some points (T6a, T11a and T12a) are shifted towards the part of the plane characterized by positive values of a*, where the red component predominates over the green one (Figure 7c). However, the relative differences $\Delta a^*$ calculated between the reference uncolonized granite and the color points from Surface a at $t_1$ (Table 3) show that all the data (absolute value) are noticeably below the value of 3 CIELAB units, considered the upper limit of rigorous color tolerance or noticeable change in color [61,62]. This evidence confirms that even the data show a wider distribution along the a*-axis compared to the initial situation ($t_0$), that the values of a* from Surface a at $t_1$ are similar to the ones of the uncolonized granite and that the chromatic differences with regard to the red-green components cannot be perceived by the human eye (Figure 7c).

However, in general, the $\Delta a^* < 0$ obtained for most of the color points means that the color of the uncolonized granite appears "greener" than the overall color of Surface a, except for T13a, T14a and T15a, characterized by $\Delta a^* > 0$.

Otherwise, the graph in Figure 7c highlights at $t_1$ a general shift towards more positive values of b*, as well as a narrower distribution along the b*-axis (min = 8, max = 13.8 CIELAB units) compared to the initial situation (min = 5.3, max = 9.4 CIELAB units).

This means an increase in the yellow component on the blue one but, more relevantly, a general shifting towards the b* of the reference value of the uncolonized granite (b* = 14.6). Conversely to the a* parameter, the calculated $\Delta b^*$ between the reference uncolonized granite and the color points from Surface a at $t_1$ (Table 3) suggests that, even after cleaning, chromatic differences with regard to the yellow-blue parameter can still be observed, since most of the values exceed the limit value of 3 CIELAB units. This is not true just for three

points, T14a, T15a and T16a, corresponding to the treatments containing the three active principles, the hydrogel alone and the mechanical brushing, respectively.

Indeed, as is evident from the graphical representation, and confirmed by the values obtained from $\Delta a^*$ and $\Delta b^*$, these are the three treatments that returned the better results, since the color of the corresponding surfaces is back to appearing very similar to the one of the uncolonized granite.

## Before Cleaning ($t_0$)

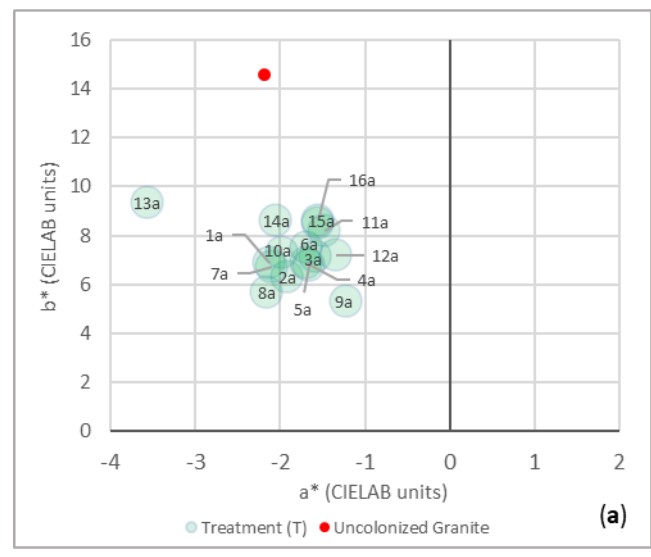

(a)

## After Cleaning ($t_1$)

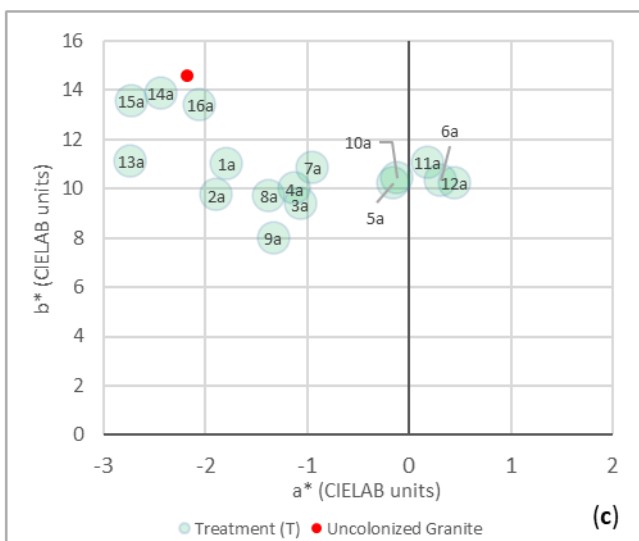

(c)

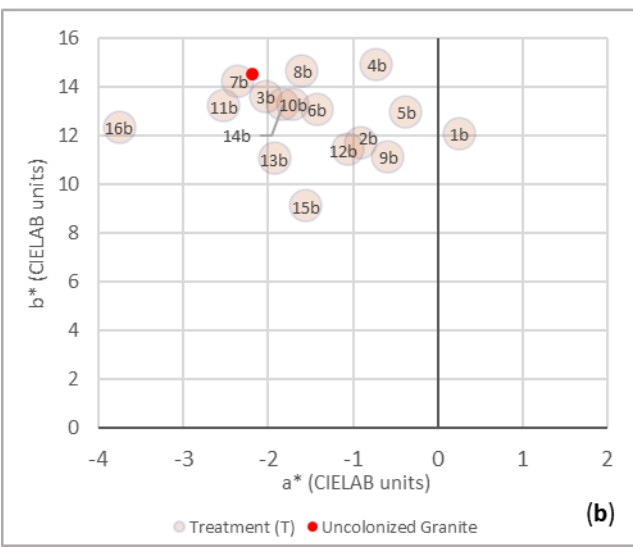

(b)

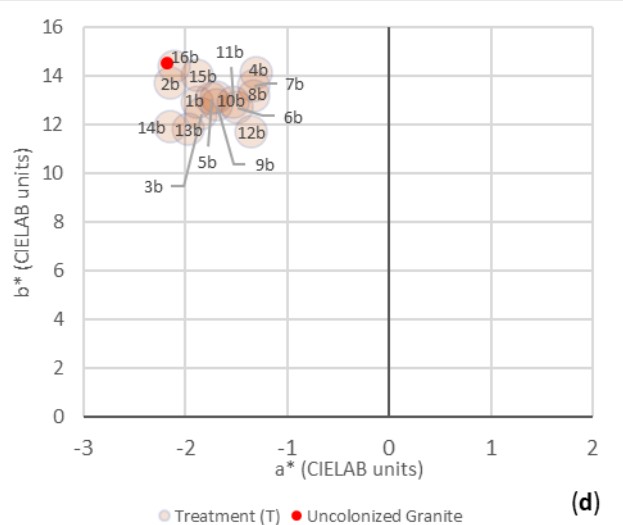

(d)

**Figure 7.** Color changes of the samples from Surface a and Surface b, before ($t_0$) and after ($t_1$) the treatment's removal, are reported in the colorimetric plane of a* (x-axis, changes in redness-greenness) and b* (y-axis, changes in yellowness-blueness). (**a**) Color data from Surface a at $t_0$; (**b**) color data from Surface b at $t_0$; (**c**) color data form Surface a at $t_1$; (**d**) color data from Surface b at $t_1$. In each graph, the color of the reference value for the uncolonized granite was also represented (red spot). Each number represents the corresponding treatment (codes are shown in Figure 3), and the letters (a and b) correspond to the surface where the treatments have been applied.

**Table 3.** Variations in the green-red (Δa*) and blue-yellow (Δb*) chromatic components, calculated between the uncolonized granite and the values from Surface a and Surface b after the cleaning procedures (t$_1$).

| Treatments | Composition | Surface a | | Surface b | |
|---|---|---|---|---|---|
| | | **Δa*** | **Δb*** | **Δa*** | **Δb*** |
| T1 | *O. vulgare* | −0.4 | 3.6 | −0.3 | 1.7 |
| T2 | *T. vulgaris* | −0.3 | 4.8 | 0.0 | 0.9 |
| T3 | *C. nepeta* | −1.1 | 5.2 | −0.3 | 2.1 |
| T4 | Carvacrol | −1.1 | 4.6 | −0.9 | 0.5 |
| T5 | Thymol | −2.0 | 4.4 | −0.4 | 1.5 |
| T6 | Pulegone | −2.5 | 4.2 | −0.7 | 1.9 |
| T7 | *O. vulgare + T. vulgaris* | −1.2 | 3.7 | −0.9 | 1.0 |
| T8 | *O. vulgare + C. nepeta* | −0.8 | 4.9 | −0.9 | 1.4 |
| T9 | *C. nepeta + T. vulgaris* | −0.9 | 6.6 | −0.5 | 1.7 |
| T10 | Three essential oils | −2.1 | 4.1 | −0.5 | 1.4 |
| T11 | Carvacrol + Thymol | −2.4 | 3.5 | −0.7 | 1.6 |
| T12 | Carvacrol + Pulegone | −2.6 | 4.3 | −0.8 | 2.9 |
| T13 | Pulegone + Thymol | 0.5 | 3.5 | −0.2 | 2.8 |
| T14 | Three active principles | 0.3 | 0.7 | 0.0 | 2.7 |
| T15 | Hydrogel | 0.5 | 1.0 | −0.3 | 0.5 |
| T16 | Brushing | −0.1 | 1.2 | −0.1 | 0.2 |

Considering only the b* parameter, which is the one that produced the higher variations between the uncolonized granite and the treated samples, the points situated further than the one representing the color of the uncolonized granite are T9a (*C. nepeta + T. vulgaris*), T3a (*C. nepeta*) and T8a (*O. vulgare + C. nepeta*). It must be highlighted that all these three treatments contain the essential oil from *C. nepeta*.

Analyzing the results obtained for Surface b at t$_1$, it can be said that the treatments produced a better cleaning effect compared to Surface a.

This evidence is confirmed by the graphical representation of the color points inside the chart (Figure 7d) and by the calculated Δa* and Δb*(Table 3), both characterized by values (absolute value) lower than 3 CIELAB units, especially for Δa* where some points (T2b and T14b) even present a difference between a*(colonized granite–treated surface) equal to 0. In general, the minus sign for all the Δa* and the plus sign for all the Δb* establish that the color of the granite appears "greener" and "yellower" than the overall color of Surface b, although, as reported before, these differences cannot be perceived by the human eye. Such element can be also observed in the graph in Figure 7d, where all the points are very close to the reference value of the uncolonized granite.

By comparing the graphical representations at t$_0$ and at t$_1$ (Figure 7b,d), the color points at t$_1$ remarkably reduce the displacement along the a*-axis, and they are grouped in a smaller color gamut included between −1.3 and -2.1 CIELAB units for a*, and between 14.4 and 11.7 CIELAB units for b*.

Even in this case, between the most effective treatments, there are T15b and T16b, which are the only ones where the biocidal agents have not been used, but only hydrogel alone and mechanical brushing, confirming the naked-eye observations.

With regard to the treatments loaded with the EOs and the APs, the better results were obtained for T2b, containing *T. vulgaris*.

### 3.4. Inhibition of the Vital Activity of Microorganisms

The viability of microorganisms was assessed before ($t_0$) and after ($t_1$) the application of the treatments through the maximum quantum yield (Y). The graphical representations in Figure 8 report the relative differences ($\Delta Y$) obtained for Y ($t_1 - t_0$) from Surface a (Figure 8a) and Surface b (Figure 8b). Observing the results from Surface a, nine treatments out of sixteen yielded a decrease of $\Delta Y$ after the application of the cleaning procedures. For the experimental areas handled with T1a, T2a, T4a, T7a, T8a, T9a, T10a, T11a and T13a, it can be stated that cleaning produced both the removal of superficial biopatina, but also reduced vital activity of the remaining microorganisms. In particular, the treatments showing the higher decrease of $\Delta Y$ are T7a, T1a and T10a, in order of effectiveness. Each of them, in their composition, contain *O. vulgare* (T7a = *O. vulgare* + *T. vulgaris*, T1a = *O. vulgare* and T10a = *O. vulgare* + *T. vulgaris* + *C. nepeta*). The others are composed by carvacrol + thymol (T11a), pulegone + thymol (T13a), *O. vulgare* + *C. nepeta* (T8a), *T. vulgaris* (T2a), carvacrol (T4a) and *C. nepeta* + *T. vulgaris* (T9a).

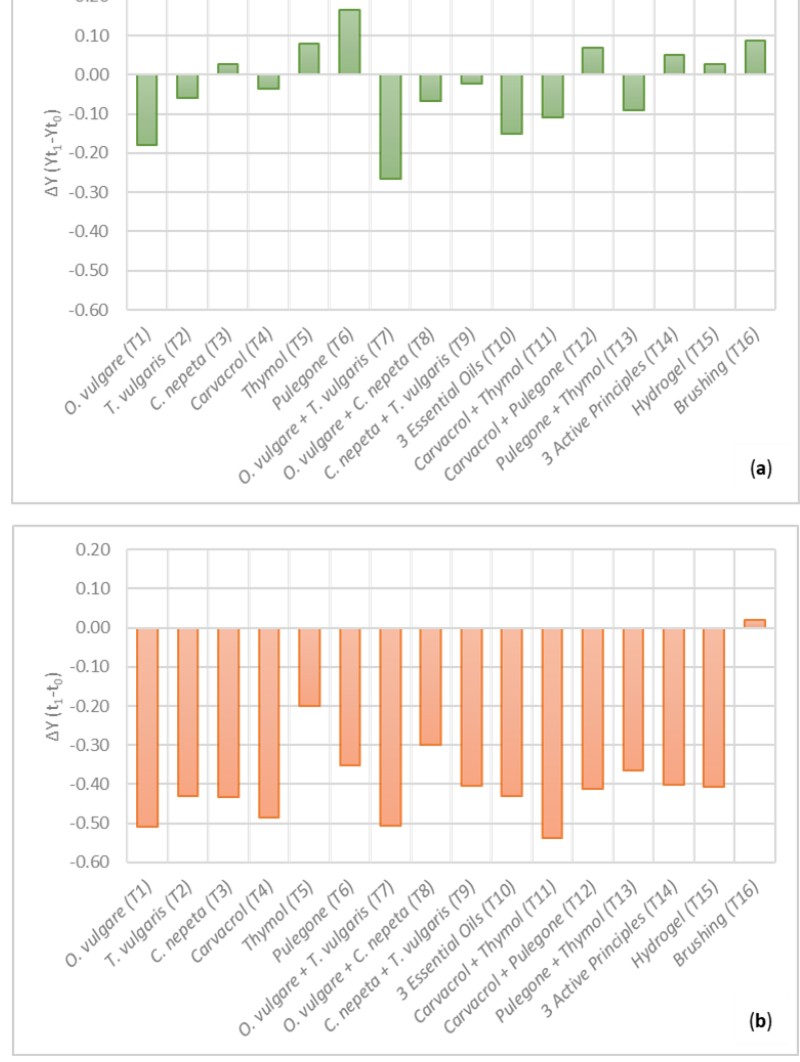

**Figure 8.** Relative differences ($\Delta Y$) of maximum quantum yield (Y) calculated between $t_0$ and $t_1$ for (**a**) Surface a, and (**b**) Surface b. $\Delta Y > 0$ establishes the increase in the vital activity after the application of the treatments; $\Delta Y < 0$ is the decrease in the vital activity.

The remaining seven treatments do not show similar results. This is the case of T3a, T5a, T6a, T12a, T14a, T15a and T16a. Between them, T3a (*C. nepeta*), T5a (thymol), T6a

(pulegone), T12a (carvacrol + pulegone) and T14a (carvacrol + thymol + pulegone) are the ones obtained by the combination of the hydrogel and the natural-based biocides. In particular, except for T3a, all of them contain the APs from the essential oils. The highest increase of ΔY is observed for the treatment containing pulegone (T6a).

On the contrary, T15a and T16a only provided mechanical removal through the hydrogel alone and the brush.

It must be noted that T14a, T15a and T16a were also the areas that, after the removal of the corresponding treatment, returned the best results in the colorimetric analysis. This is further evidence that, in this case, the removal of the superficial biopatina, intended as a "cleaning" effect, is not necessarily related to the reduction of the vital activity of the remaining organisms. This result, indeed, is expected for T15a and T16a, which are the only treatments that did not provide the employment of a substance with biological activities. At the same time, for the biofilm growth on Surface a, it was demonstrated that the effect of the APs produced, in general, worse results than the treatments containing the EOs. For this reason, it is not unexpected that the treatment based on the blend of the three APs does not produce a decrease in the viability of microorganisms either. The efficacy in the cleaning can be justified with a superior adhesion of this treatment on the surface and a consequent effective removal of the biopatina that contains not only the microorganisms, but also the EPS and other substances that can be entrapped in the biological matrix.

On the other hand, the same treatments applied on Surface b produced different results (Figure 8b). The only treatment that showed an increase of ΔY after the cleaning procedures is the mechanical brushing (T16b). Even in this case, although the colorimetric results confirmed an efficient removal of the superficial biopatina, the absence of a biocide in the composition did not produce a decrease in their viability. Otherwise, a decrease of ΔY is recorded for all the other treatments, and the absolute values are much bigger than the ones obtained for Surface a. This confirms that the biofilm from Surface b responded much better to the application of the treatments loaded with the phyto-based substances than the biofilm from Surface a, but also to the hydrogel (T15b), which proved to be an efficient tool for the complete removal of the superficial layers, as already demonstrated in other studies [28,29]. The treatments that produced the higher decrease of ΔY are T11b (carvacrol + thymol), T1b (*O. vulgare*), T7b (*O. vulgare* + *T. vulgaris*) and T4b (carvacrol). All of them contain *O. vulgare* or its AP (i.e., carvacrol), in their composition. Furthermore, it must be noted that T1b contains the same concentration in carvacrol as T4b, and T7b contains the same concentration in carvacrol and thymol as T11b, evidencing a correspondence between the treatments based on the essential oils with the ones containing the active principles.

They are followed by *C. nepeta* (T3b), *O. vulgare* + *T. vulgaris* + *C. nepeta* (T10b), *T. vulgaris* (T2b), carvacrol + pulegone (T12b), hydrogel (T15b), *C. nepeta* + *T. vulgaris* (T9b), carvacrol + thymol + pulegone (T14b), pulegone + thymol (T13b), pulegone (T6b), *O. vulgare* + *C. nepeta* (T8b) and thymol (T5b), which is the treatment that produced the lower decrease of ΔY, although it is still higher than most of the data collected from Surface a.

## 4. Discussion

Essential oils have assessed biocidal properties, attributed to their interaction with a microorganism's cellular membrane [13], although the substances involved and the mechanisms occurring must be investigated in depth. In this regard, the presence of some chemicals in the composition of EOs seems to improve their biocidal and fungicidal effects. This was demonstrated in many studies, among others that of Mironescu et al. (2010) [63], where some essential oils from Lamiaceae (included *Thymus vulgaris*) characterized by a predominance of monoterpenoids (carvacrol, thymol and estragol) showed stronger antifungal properties than other EOs mainly composed of hydrocarbons. In the same study, it was also demonstrated that the biocidal properties of each EO is higher or lower depending on the microorganism involved.

All these elements justify the increased interest of the scientific community in the employment of essential oils as biocides for the treatment of biocolonization on cultural

heritage, also in virtue of their recognized lower environmental impact compared to chemical biocides. However, only a few studies have provided similar evaluations taking only into account the active principles of the EOs [18,19,64], and the argument needs further insights.

In light of the above, this study aimed to assess the potentialities of three selected essential oils rich in phenolic monoterpenes and already known for their biological activities (*O. vulgare*, *T. vulgaris* and *C. nepeta*) and their APs (carvacrol, thymol and R-(+)-pulegone), used either alone and blended in combination of two or three compounds, to be employed on cultural heritage materials, as a possible alternative to classical biocidal methods. Indeed, this study is in continuity with two previous works, where a protocol for the application of such substances on stone materials was designed by loading them in a gellan PVA-based hydrogel and monitoring their effect through non-invasive colorimetric measurements. In those cases, the applications were performed in a laboratory on artificially biocolonized granite samples [29] and on site on a spontaneous biofilm growth on travertine [28]. As an advancement, here we propose systematically evaluating in situ the effect of the previously cited substances against the growth of two phototropic biofilms on a granite wall, with a specific focus on the effect of the phyto-derivatives towards the different microbial species composing the biofilms, considering the following: (i) the contribution of the APs to the biocidal properties of the Eos; (ii) an eventual synergic effect associated with the combination of several compounds; and (iii) the suitability of cleaning methods, evaluating the differences occurring between the two biofilms. Indeed, we also characterized the EPS produced by the microorganisms as a further indicator of the healthy status of the biofilms and their eventual resistance to removal.

Furthermore, the strategy for the assessment of the cleaning and biocidal action was improved by pairing colorimetric investigations with the determination of the viability of microorganisms through chlorophyll *a* fluorescence measurement. Indeed, these are two complementary tools, since colorimetry provides important information about the aesthetic impact of biofilms on surfaces, but chlorophyll *a* fluorescence's detection is more useful in estimating eventually remaining phototrophic biomass [65] and thus a possible persistence of the treatments over time. A similar approach was already employed in several studies dealing with the treatment of biocolonization on stone materials [5,48,50,65–67]; more recently, in other studies providing similar applications of EO-based systems on biocolonized stone-made artifacts, the suitability of such non-invasive methods was also demonstrated [27,30,31], and the creation of a systematic protocol for this kind of application on cultural heritage is required.

Firstly, the morphological and taxonomic identification confirmed the different composition of the two biofilms colonizing Surface a and Surface b, where the first one was characterized by co-dominance of algae (mainly *A. Lobatus*) and cyanobacteria (mainly *Nostoc* sp. and *Gloecapsa* sp.), and the second showed a predominance of green algae (mainly *T. aurea*). The difference in the microbiological composition, leading to the production of specific photosynthetic pigments, was the cause of the different chromatic appearances of the two biofilms [68]. Differences in color detected from the surfaces were also univocally determined by colorimetric investigations, where the chromatic parameters a* and b* and their variations ($\Delta$a* and $\Delta$b*) were considered for analytical purposes, since they proved to be useful tools for the early detection of photosynthetic biofilms [47], as well as for the evaluation of cleaning efficacy of restoration products against photosynthetic biofilm growth on granite-built historical surfaces [5].

As expected, the different microbial compositions produced a different response to the application of the treatments. The higher content of EPS produced by microorganisms from the biofilm characterized by a co-dominance of algae and cyanobacteria (Surface a) justify the fact that the cleaning procedures gave worse results in this case, compared to the efficient removal of the superficial biofilm characterized by a predominance of green algae (Surface b), as detected by colorimetric investigations and by the determination of the viability of the remaining microorganisms. In fact, a large content of EPS provides protection and

cohesion of biofilms and stronger adhesion of microorganisms to the substrata, giving them higher resistance to the access of toxic compounds to the cells [69,70], as occurred in this case after the application of the treatments. In such circumstances, a repeated application of the treatments can be required to obtain a complete removal of the biopatina, as performed in other cases [31].

Considering the cleaning effect alone, pure hydrogel and mechanical brushing are the most effective treatments for biopatina removal, as observed both for Surface a and Surface b. These results are consistent with those obtained in our previous study performed on biocolonized granite samples [29], where the same hydrogel and mechanical brushing were used. Also in that case, mechanical brushing efficiently removed the superficial biopatina, although colorimetric measures demonstrated that it was the only treatment that did not produce chromatic variations in the color of uncolonized granite (used as a reference) over time. Such evidence can be explained, since the mechanical method is useful for eliminating the superficial dirty layers from surfaces but does not ensure a decrease in the vital activity of remaining microorganisms (as evaluated in this case for both the surfaces by the PAM-fluorescence results) and does not prevent a future recolonization [71]. Furthermore, the employment of mechanical tools, eventually coupled with a biocide, is not recommended for applications on cultural heritage, given their abrasive and aggressive properties [5]. At the same time, the hydrogel already demonstrated its suitability in the efficient removal of superficial patinas in both of the previously cited studies, where it seems to consistently reduce the biomass [28,29]

In terms of natural biocides, the treatments containing *O. vulgare* were the most effective against both the biofilms. This result is in agreement with other studies [69] that already demonstrated the broad biocidal action spectrum of such compound towards several microorganisms, such as fungi, bacteria, cyanobacteria and algae, found on various cultural heritage materials [54,72–74]. As an example, a study of Argyri et al. (2021) [75] tested eighteen high-quality essential oils from Greece, including extracts from *Salvia* sp., *Foeniculum* sp., *Satureja* sp, *Juniperus* sp., *Citrus* sp., *Laurus* sp. and *Origanum* sp., against 35 bacterial and 31 fungi isolated from a Greek cave, and the better results were obtained for both the EOs from *O. vulgare*, which inhibited the growth of all the strains at very low concentrations (0.1% *v/v*).

In this study, the presence of *O. vulgare* seems to have a synergic action when combined with *T. vulgaris*, since the blend of these EOs was the most effective out of the treatments containing two or three EOs. This is verified for both the biofilms studied and is in accordance with similar results obtained in our previous research towards a biofilm where species from fungi, bacteria and algae were identified [28], and in a pilot work, coordinated by the Vatican Museums, where the effectiveness of a mixture of such compounds, combined with *Funori*, showed its effectiveness in removing a phototropic biofilm (*Chloroccoccum* sp., *Chlorella* sp., *Nostoc* sp. and *Phormidium* sp) from statues preserved in the Vatican Gardens [59]. Indeed, another recent work of Spada et al. (2021) [30] confirms that treatments providing the blending of *O. vulgare* with other EOs showed their effectiveness against a complex biofilm containing various species of cyanobacteria (*Phormidium* sp., *Calothrix* sp., *Chroococcus* sp., *Gloeocapsa* sp) and they can be considered a reliable alternative to classical biocides, as much as a mixture containing common oregano and other EOs was selected for the treatment of a real study case (Statue of Silvanus, National Archaeological Museum of Florence) [31]. In the same study, it was assessed that the EOs from *T. vulgaris* showed contradictory results [13,30], and this assertion was also matched in this study when *T. vulgaris* is not coupled with *O. vulgare*.

Furthermore, a study by Gagliano Candela et al. (2019) [21] evidenced that the high potential of the EOs from *Thymbra capitata,* for the inhibition and elimination of biological patinas of cyanobacteria and green algae from three outdoor surfaces (ceramic, marble and cement grit), must be attributed to the high concentration of carvacrol, but also to the minor contribution of γ-terpinene and p-cymene, enabling carvacrol to be more easily transported into the cells [76].

Indeed, γ-terpinene and p-cymene were also detected in our oils from *O. vulgare* and *T. vulgaris*, in non-negligible percentages [29], while carvacrol is considerably more abundant in *O. vulgare* (70.5% of EO composition) than in *T. vulgaris* (3.8% of EO composition).

Thus, carvacrol is the main factor responsible for the biocidal action, also considering the results obtained in this work, where carvacrol alone, combined with other APs (especially thymol) or as a main component of an EO (*O. vulgare*), can be considered effective against two complex sub-aerial biofilms.

However, the combined presence of carvacrol with other compounds, as naturally happens in essential oils, seems to improve its effect and at the same time justifies the poor results obtained in some cases for the biofilm characterized by a co-dominance of green algae and cyanobacteria (Surface a) treated with APs alone. On the other hand, towards the biofilm from Surface b (mainly green algae), the biocidal properties of APs are strong enough to produce a reduction in the microorganism's viability, comparable or higher than the effect produced by EOs, always taking into account the different composition of the biofilms and their adhesion on substrates. The different behavior observed for Surface a and Surface b for the same treatments confirms that the efficiency is strongly related to the interaction between APs (alone or in a mixture, i.e., in EOs) and the specific composition of the biofilm. As further confirmation, in our previously cited study [29], where the same treatments applied therein were used on a different colonization, *C. nepeta* and its main component R-(+)-pulegone, proved to be the most effective against a biofilm composed of *Bracteacoccus minor*, *Stichococcus bacillaris*, *Chlorella* sp., *Isocystis* sp., *Aphanocapsa* sp. and *Leptolyngbya cebennensis*. *C. nepeta* indeed also demonstrated very effective results in other works [35], conversely to this case, especially against the microorganisms from Surface a, where it did not show the same efficacy.

For these reasons, although EOs and their APs must be considered a valid eco-sustainable alternative to classical biocides, it must be always considered that their biocidal action is strictly related to a biofilm's composition and further investigation is required, especially concerning the pioneer phototropic microorganisms (algae and cyanobacteria), on which the effect of such substances has been little studied compared to fungi and bacteria [13].

Actually, this aspect is encouraging, since it appears evident that it will be possible to create products with various compositions designed for specific target microorganisms.

Such formulations can be produced both considering only the Eos and, more conveniently, the single APs properly combined.

The encapsulation of these compounds must be always provided for applications on cultural heritage, in order to reduce the volatility of phyto-compounds. Furthermore, the presence of phyto-derivative agents not only confers biocidal properties to the carrier, but also modifies its rheological properties, and the hydrogel becomes less adhesive to the surfaces and thus less aggressive, preventing the undesirable complete removal of the patina, in compliance with the restoration theories from Cesare Brandi [77].

Finally, it must be remembered that EOs from *T. vulgaris* and *O. vulgare* applied to the construction materials from the Roman wall of Lugo (NW Spain, UNESCO site) has not produced any alteration on the composition or the aesthetic appearance of the treated materials years later [78], providing further proof of their stability over time as restoration products.

## 5. Conclusions

This study confirms the potentialities of hydrogel systems loaded with phyto-derivatives for the treatment of biocolonized cultural heritage, as a promising alternative to classical biocidal methods.

Essential oils showed, in some cases, broader biocidal action towards two complex phototropic biofilms, although the results obtained for the active principles are still satisfactory. Indeed, an eventual employment of the APs alone in the same concentrations present in the essential oils has several advantages, among which is the possibility of systematically

approaching the realization of bio-inspired biocidal formulations at fixed concentrations, active against specific microorganisms. In addition, the economic advantage derived cannot be ignored, since the extraction of high-quality essential oils is quite expensive and requires a large amount of vegetal material to obtain small quantities of extracts.

Overall, the EO from *O. vulgare* proved to be the most effective substance, considering the good results obtained against both biofilms. The high percentage of carvacrol seems to be the main factor responsible in *O. vulgare*'s biological activities, although, against more resistant biofilms, the presence of other substances naturally present in the oil increases carvacrol performance. Furthermore, when *O. vulgare* is combined with determined EOs, especially *T. vulgaris*, a synergic action can be observed. This is also true when considering similar results obtained by some mixtures only containing the APs applied on both biofilms, especially when carvacrol is combined with thymol (respectively, the APs of *O. vulgare* and *T. vulgaris*). For this reason, the effect of the substances alone towards specific microorganisms must be always studied on a case-by-case basis depending on the biofilm's composition, in order to create products ad hoc for a precise reduction of characteristic colonies of targeted species.

The analytical approach included the employment of portable instrumental tools, and colorimetric investigations, combined with the assessment of the chlorophyll *a*, proved to be two complementary and effective strategies for the assessment of the cleaning and biocidal effect of the substances. The complete non-invasiveness of such methods has as a main advantage the possibility of repeating the analyses without adding any damage to the study cases, and it is particularly useful for applications on cultural heritage materials. In light of the above, the future insights of this research will provide long-term monitoring in order to assess a possible persistence of the compounds on the surfaces, considering both the prolonged biocidal action and, on the other hand, eventual adverse variations of the surfaces due to the aging of the products, in order to exclude any secondary unwanted effects for future applications on cultural heritage.

**Author Contributions:** Conceptualization, C.G., E.F. and B.P.; methodology, C.G., E.F. and B.P.; software, C.G. and E.F.; validation, C.G., E.F. and B.P.; formal analysis, C.G. and E.F.; investigation, C.G. and E.F.; resources, B.P.; data curation, C.G. and E.F.; writing—original draft preparation, C.G., E.F., G.F. and B.P.; writing—review and editing, G.F. and B.P.; visualization, G.F. and B.P.; supervision, B.P.; project administration, G.F. and B.P.; funding acquisition, G.F. and B.P. All authors have read and agreed to the published version of the manuscript.

**Funding:** This research was partly financed through project ED431C 2022/09 (Xunta de Galicia). E.F. was financially supported by a PhD Fellowship-Contract MICINN-FPI (BES-2017-079927).

**Institutional Review Board Statement:** Not applicable.

**Informed Consent Statement:** Not applicable.

**Data Availability Statement:** All data that support the findings of this study are included within the article.

**Acknowledgments:** The authors are grateful to P. Matricardi (Sapienza University of Rome, Rome, Italy) and C. Cencetti (QI technologies s.r.l, Pomezia (RM), Italy) for the ideation of the hydrogel and QI technologies s.r.l. for making available the hydrogel itself and the materials necessary for its realization.

**Conflicts of Interest:** The authors declare no conflict of interest.

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
