# Peer review of "Evaluation of the Cleaning Effect of Natural-Based Biocides: Application on Different Phototropic Biofilms Colonizing the Same Granite Wall"

_coatings, doi:10.3390/coatings13030520_

Round 1

Reviewer 1 Report

This manuscript discusses the evaluation of different natural products, including essential oils from ganum vulgare, Thymus vulgaris and Calamintha nepeta, to remove two heterogeneous superficial biopatinas from a naturally biocolonized granite wall. The study aims to assess the effectiveness of these natural compounds in removing biopatinas and their potential synergistic effects on specific microorganisms. The article also compares the cleaning properties of the natural compounds with a hydrogel and a mechanical method used as controls. The authors used colorimetry and PAM fluorometry investigations to assess the effect of the treatments in removing biopatinas and the vital activity of microorganisms that survived after the experimental procedures. The study concludes that the natural compounds showed promising results in removing biopatinas, and further experiments are necessary to evaluate their interference with heritage materials.

The authors have reached the original results. However, I have these comments on the manuscript.

The current issues are summarized below:

1.      Section 2.2: Experimental setup, please consider removing the details discussion from Page 5 (line 177-197) and include it in to result and discussion.

2.      The caption for Figure 5 should be more concise.

3.      Section 3.3, please combine the first three paragraphs as one paragraph.

4.      Please combine the following two paragraphs:

Page 14 (line 464):

On the contrary, T15a and T16a only provided the mechanical removal through the 464 hydrogel alone and the brush

The following paragraph:Line 466-478

5.      The Results and Discussion section needs more comparisons from the literature.

Reviewer 2 Report

The study was well-designed, and the results are detailed enough to examine the mechanisms behind the reported trends. There is a need to justify the following comments, in order to understand their practicability for the stated objectives.

Comments to the Author

The abstract is too long. Please concise the abstract and only provide the main findings.

The introduction should include research justification that led to the three Essential Oils (EOs) from Origanum vulgare, Thymus vulgaris, and Calamintha nepeta.

The aims of the study are not clear. Please further elaborate on the aims and give more details.

The results and discussion do not go further than expected. It lacks a deepening. The results need proper discussion and comparison with previous similar studies to highlight the usefulness of the study.

The novelty of the manuscript is questionable as already a lot of literature has been published. Please highlight the novelty of the manuscript

The conclusion needs to be improved. The conclusion is too general. What were the outcomes and prospects are not clear? The authors should justify elaborately how the current study is different from previous studies and useful to the scientific community.

Reviewer 3 Report

Dear Authors

Your study is very interesting. It is well organized and contains the most critical parts of a scientific study.

Namely, the protection of granite walls in terms of biofilm reduction is very important in everyday life and for the protection of historical sites; restoration products, and taking care of biological patinas.

The application of compounds with natural origin like essential oils as new biocidal substances is a privilege for this study. They are proven that they could be used for this purpose due to the wide spectra of chemically active substances that they contain and at the same time they are environmentally safe.

Therefore, the objective of the study is well established. The introduction part provides a good overview of the background of the topic. The methodology of the experimental part is also well organized and the critical points of the study are considered. Consequently, the results and discussion sections follow the experimental part and present the obtained data in a clear and easily readable way. 

The Conclusion is given in a short and brief way, covering the most important results of the study.

Please find the attached version of the manuscript where some comments and suggestions are given in order to improve the article.

I hope that this will contribute to a better version of this manuscript.

Thank you.

Author Response

The authors are very grateful to the Reviewer for her/his positive comments on the manuscript. We provided to accomplish all the suggested modifications, that are visible in the attached PDF where we highlighted the made changes through the "Track Changes" function of word, as well as the replies to each comment made by the rewiever.

Thank you very much

Reviewer 4 Report

Comment:

Overall, I enjoyed reading this article which was written clearly and conscientiously. Having said that, in my opinion, this manuscript can be recommended for publication in such a reputable scientific journal after minor revision:

  1) Abstract: Essential Oils, Active Principles SHOULD BE essential oils, active principles. Please change that throughout the text.

Author Response

The authors are grateful to the Reviewer for her/his positive comments on the manuscript. We provided to accomplish all the suggested modifications on the manuscript.

Round 2

Reviewer 1 Report

The author has made the necessary revisions to the manuscript, and I am satisfied that it now meets the standards for acceptance.